# Minimizing The Number of Channel Switches of Mobile Users in Cognitive Radio Ad-Hoc Networks

**Rajorshi Biswas *** and **Jie Wu**

Department of Computer and Information Sciences, Temple University, Philadelphia, PA 19122, USA; jiewu@temple.edu
* Correspondence: rajorshi@temple.edu

**Abstract:** Cognitive radio (CR) technology is envisioned to use wireless spectrum opportunistically when the primary user (PU) is not using it. In cognitive radio ad-hoc networks (CRAHNs), the mobile users form a distributed multi-hop network using the unused spectrum. The qualities of the channels are different in different locations. When a user moves from one place to another, it needs to switch the channel to maintain the quality-of-service (QoS) required by different applications. The QoS of a channel depends on the amount of usage. A user can select the channels that meet the QoS requirement during its movement. In this paper, we study the mobility patterns of users, predict their next locations and probabilities to move there based on its history. We extract the mobility patterns from each user's location history and match the recent trajectory with the patterns to find future locations. We construct a spectrum database using Wi-Fi access point location data and the free space path loss formula. We propose a machine learning-based mechanism to predict spectrum status of some missing locations in the spectrum database. We formulate a problem to select the current channel in order to minimize the total number of channel switches during a certain number of next moves of a user. We conduct an extensive simulation combining real and synthetic datasets to support our model.

**Keywords:** cognitive radio networks; channel hand-off; mobility pattern; spectrum database

---

## 1. Introduction

In the cognitive radio network (CRN) architecture, the users use the unused spectrum for their communication. There are two types of communication: communication between a user and a cognitive radio base station and communication between two users. Cognitive radio (CR) users create connections among them and a local ad-hoc network. The ad-hoc network is important for local file sharing, call, or message. Different types of applications need different data rates. Each CR user has a minimum data rate which depends on their applications. The achievable data rate in a channel depends on many channel characteristics, including signal power, signal-to-noise ratio (SNR), and the number of users. These characteristics are different for different channels in different locations. The CR users must be guaranteed with the minimum required data rate during their movement to a different location. To ensure the minimum required data rate, a CR user needs to switch to different channels while it is moving. Channel switching is a costly process because it takes some time to re-establish the connection. Finding a suitable channel is also a time-consuming process. Channel selection is hard when multiple channels offer higher data rates than the required data rate.

For example, in Figure 1 there are three unlicensed channels $CH_1$, $CH_2$, and $CH_3$. At block 1, the overall received signal power from different Wi-Fi access points (APs) in $CH_1$, $CH_2$, and $CH_3$ are −90 dBm, −70 dBm, and −90 dBm, respectively. In general, signal power higher than −30 dBm is considered as the best signal, −30 dBm to −60 dBm as good, and −60 dBm to −80 dBm as poor signal.

For this example, we consider signal power $-90$ dBm or less than that as environmental noise. At time $t_0$ the user is located at block 1. In this block, there are two unused channels which are $CH_1$ and $CH_3$. Let us assume the user selects $CH_1$. At time $t_1$, the user moves to block 2 where only $CH_1$ is being used. Therefore, the user switches to $CH_2$. When the user moves to block 4, it also needs to switch to $CH_3$ from $CH_2$. In this option, the total number of channel switches is two. If the user selects $CH_3$ at time $t_0$ in block 1, then it does not need switch channels during its movement from block 1 to 4.

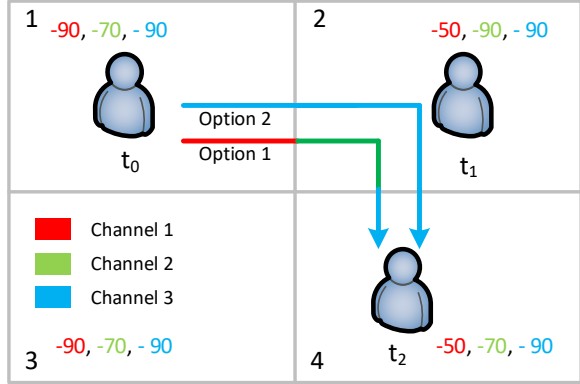

**Figure 1.** User mobility and channel hand-off.

In this paper, we aim at minimizing the number of channel switches of the CR users by ensuring the minimum required channel quality of each user during its movement. To achieve the minimum number of channel switches over a period, we need two things: a spectrum database and a mobility predictor. The spectrum database contains channel state information of all locations. The mobility predictor predicts the next possible location of a user by using the location history of CR users. To the best of our knowledge, there is no public spectrum database that considers all unlicensed bands.

Minimizing the number of channel switches is challenging because of the following reasons. Firstly, there is no public spectrum database. We consider the 2.4 GHz and 5 GHz unlicensed bands for CRN operations. There is some public database including Google spectrum databases that keeps the spectrum usage information of TV bands (54–890 MHz). A TV tower covers an area with a radius ranging from 20 miles to 120 miles. Therefore, in a city, the TV spectrum usage remains similar. A CR user using TV white space does not need any spectrum switch while it is moving within a city. On the other hand, the devices that operate on 2.4 GHz and 5GHz have transmission range from 50 m to 100 m. Therefore, spectrum usage on these bands is different in locations within a small distance.

Secondly, there are some public mobility datasets including Microsoft Geolife GPS trajectory dataset [1]. The locations of the users are mainly at Beijing, China. It is necessary to have both datasets at the same location. Shifting the locations of a dataset might not be realistic because the movement depends on many factors including infrastructure, roadmap, and population. Therefore, it is challenging to port mobility data from one location to another.

To overcome these challenges, we need to construct a spectrum database. We can collect channel states and construct a spectrum database on a small scale. We divide the experiment area into small blocks (for example $50 \times 50$ m block). Each block is sensed once in the experiment area. In some cases, it might not be possible to sense every block. We present a spectrum usage predictor that can predict the spectrum usage at the unseen block by using the information of the surrounding blocks. To do a large scale evaluation we construct a synthetic spectrum database from the AP location dataset. Then, we use pattern matching-based approach for predicting future locations of a user. Finally, we formulate an optimization program to find the channel assignment of a user for future locations. The problem transforms the shortest path problem and the solution is found using graph search-based algorithms. The main contributions in this paper are the following:

- We propose a mechanism to fill up the channel status information at missing locations in the spectrum database using some machine learning techniques.

- We formulate and solve a problem to minimize the number of channel switches by predicting the future locations of users based on their location history.
- We conduct extensive simulation with a real and a synthetic dataset to support our model.

The remainder of the paper is as follows. Section 2 describes some related works on constructing a spectrum database, finding future locations of users, and minimizing the number of channel switches. Section 3 presents the basic information on spectrum and cognitive radio ad-hoc network system. Section 4 describes the system model we consider for this work. Our future location prediction scheme is presented in Section 5. Some experiments and machine learning methods for finding missing data in the spectrum database are in Section 6. In Section 7, experimental results are presented. Finally, Section 8 concludes our paper.

## 2. Related Work

There are several existing works on mobility prediction of users, spectrum sensing, and minimization of channel switches.

Firstly, we discuss some of the recent works on constructing a spectrum database and predicting channel status. In [2], the authors design a channel status predictor using two different adaptive schemes: a neural network based on multi-layer perceptron and a hidden markov model. The proposed channel status predictor improves the spectrum sensing operation by saving the sensing energy. In [3], the authors present an efficient method to construct a white space database. The TV white space is considered only in this database. White space devices are employed to build a TV white space database. They combine the geo-location database and spectrum sensing scheme and the signal strength calculated by a signal propagation model. They divide the area into a large number of small grids to construct the TV white space database. Dividing the area makes the data processing more convenient. In [4], the authors present a method to construct a multi-dimension spectrum database. The dimensions are time, frequency, and location. They first introduce the concept of spectrum tensor to depict the multi-dimensional spectrum data. They develop a prediction scheme to predict the incomplete measurements in the spectrum database. There are some other approaches for predicting channel status and constructing the spectrum database based on different machine learning approaches [5–9].

Secondly, we discuss some of the recent works on modeling mobility patterns of users and predicting future locations using location history. In [10], the authors focus on improving the prediction accuracy of human mobility data by constructing a hybrid markov-based model. A cluster-based future location prediction system is proposed in [11]. The model uses the past trajectory data of all users to predict the next location of a user. The predictor uses non-parametric Bayesian statistical tools. In [12], the authors use filter-based techniques to predict the target and a curve-fitting algorithm to model both linear and non-linear mobility models of users. They also adjusted the efficiency of the algorithm by comparing its mobility prediction vis-a-vis the Kalman filter. In [13], the authors use the k-means and self-organizing map to cluster the mobility pattern of users. The mobility data is the connected cellular tower locations of a user at different times. In this approach, the patterns are stored in a database and matched with the current mobility information to find the next locations. In [14], the authors present a solution to the Nokia mobile data challenge. The users are not allowed to build joint models over other users. So, all the users' predictors are independent of each other. The predictor takes the current location, time, and some additional data from the user's device. Therefore, the predictor cannot access the history of users and predict based on current locations. The predictor uses a mixer of dynamic bayesian network, artificial neural networks, and gradient boosted decision trees for predicting the next location.

Finally, we discuss some existing works on minimizing the number of channel switches. In [15], the authors formulate a multiple-objective optimization problem and simulate using the cuckoo optimizing algorithm to minimize the number of handovers when a user is moving. They consider a scenario where different wireless networks are overlapping. In [16], the authors propose a smart

proactive channel scanning scheme that enables the users to perform fast handover. The works mentioned here are either focusing on building a spectrum database or predicting future locations of users. These two types of works are very important to minimize the number of channel switches in cognitive radio networks. Therefore, research is needed for minimizing the number of spectrum switches by combining these two types of research.

## 3. Overview of Cognitive Radio Ad-Hoc Network

Currently, governmental agencies assign wireless spectrum to mobile operators for a long term and large area. This kind of static spectrum allocation restricts other operators' users to use the spectrum that are assigned to other operators. This is how the licensed user gets exclusive access to the spectrum. We call the assigned spectrum to licensed spectrum and the users of that operator are called the primary users (PUs). Besides the licensed spectrum, there are some spectrums that can be used by any wireless user without having a license or exclusive access. This type of spectrums are called the unlicensed spectrums. There are some limitations of using the spectrum including power limitation and mandatory sensing before transmission. Spectrum crisis of mobile users increases day by day with the increase of data transmission. The trust for more spectrum results in development of cognitive radio architecture which uses a dynamic spectrum allocation. Users use their spectrum in an opportunistic manner. An opportunistic manner means if a PU is not using its spectrum, other users can utilize that spectrum by using it. We call the other users as secondary users (SUs). They are low priority users and must evacuate the spectrum for PU if they need it.

In this kind of network architecture, SUs need to sense channels to find the unused channels. If there are more than one unused channel they need to choose the best channel. Usually, the number of SUs is much greater than the number of unused channels. Therefore, an SU needs to share a channel with many other SUs. While using an unlicensed channel, the PU may start transmitting. If a PU transmission is detected by an SU, it must vacate the channel. So, there are four following major steps in CR networks:

- **Spectrum sensing:** Channels are sensed to find whether they are occupied by PUs or not. A channel is free if there are no licensed transmissions.
- **Spectrum decision:** After spectrum sensing an SU may have multiple free channels. In this step, SUs find one of the best free channels to transmit the data.
- **Spectrum sharing:** The number of free channels is smaller than the number of SUs. Therefore, multiple SUs share a channel.
- **Spectrum mobility:** SUs are mobile and they need to switch channel while they move to another location. The channel they are using may not be free at the new locations. Even when the SU is static, the operating channel may be occupied by the PU later. In both situations, the SU needs to vacate the channel and find another channel to continue operation.

From these steps in CR networks, we observe that there are two kinds of transmission: primary transmission which happens on a licensed channel and secondary transmission which happens on a unlicensed channel. Accurate spectrum sensing is very important because transmission in unlicensed channels depends on sensing information of the CR users. There are several methods to detect primary transmission including primary transmitter detection, primary receiver detection, and cooperative sensing. In the primary transmitter detection method, a weak signal from a primary transmitter is detected through the local observations of users. In the primary receiver detection method, local oscillator leakage power emitted by the RF front-end of the primary receiver is detected. In the cooperative sensing method, SUs makes decision about PU presence by combining sensing information of a group of SUs.

An SU must be able to choose the best channel among the free channels. Spectrum selection depends on multiple channel characteristics including the number of SUs in that channel, PU activity, path loss, interference, and contagious free channel availability. An SU must be able to share the same

channel with multiple SUs. Many SUs can select the same channel as for operation. Therefore, they need to share the same channel with different other SUs. When an SU is transmitting in an unlicensed channel, a PU may show up to use the channel and it must vacate the channel to the PU. The secondary transmission cannot be stopped and the SU must find another channel and continue transmissions on that channel.

In a cognitive radio ad-hoc network, the users directly communicate with each other without using the base station (BS). Figure 2 shows a scenario of CR network. We can see that users 1 and 2 are connected with the primary BS. Users 4, 6, and 7 are connected to the CR BS. User 3 and 5 are not connected with any of the BSs. We call the user 1 and 2 as PU. User 3, 4, 5, 6, and 7 are SUs. In this scenario, user 3, 4, 5, and 6 create an ad-hoc network. The users in the ad-hoc network can directly transmit to their neighbors. Each user also works as a router and helps to deliver messages to the users that are more than one hop away. The ad-hoc users without secondary BS connectivity can also reach the BS through multi-hop communication. When the SU moves from one place to another the structure of the topology, channel, and routing changes.

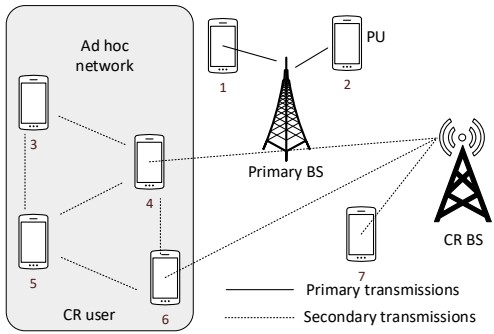

**Figure 2.** CR ad-hoc network.

## 4. System Model

We consider a CRN in 2.4 GHz and 5 GHz unlicensed channels with some SUs and always off PUs. The reason for selecting unlicensed channels is that its range is 100 m. There are a lot of APs and the channel state changes at a small distance. So, we do not need to travel a long distance for collecting channel state data. On the other hand, if we would consider the TV white space for the experiments, we would need to travel hundreds of miles for collecting the channel state. This is because a typical TV transmission travels 60 to 80 miles, depending on the height of the transmitter antenna and the transmission power. The collection of channel state is done periodically in a different time and stored in a database that is managed by CR BS. We assume that the channel state information at a particular time in a day is similar to that time in another day. Therefore, the spectrum database contains the hour, block location and signal strength of each unlicensed channel in 2.4 GHz and 5 GHz. The database is accessible to all CR users at any time.

We divide the considered area into 50 m × 50 m blocks. The reason for selecting this size is the transmission range. As the transmission range is 100m we assume that the channel state will be similar in a 50 m × 50 m block. In reality, the channel state can change in every meter. There is a trade-off between high accuracy and efficiency of calculation. If we consider the TV white space for the experiments, we would need to select the block size 50 × 50 miles for collecting the channel state.

We consider that the users move from one place to another and their location information is stored locally in their devices. A user does not share its mobility information with another user or the CR base station for privacy issues. The location information is stored block-wise. The user device runs some prediction algorithms to predict their next locations. Therefore, the prediction algorithm takes the block locations and times as input and predicts the next locations with the probability and time of moving there.

After predicting the next block locations and times of traveling to that location, the user finds the signal power in that block and selects some of the channels that are usable based on its need. It is very complex to select a channel based on signal power and the number of APs in that channel. For simplicity, we categorize the channel status in three categories: highly used, medium used, and almost free. The qualified channels are listed based on the users' application. For example, if a user is running a chat application or text messages then it needs a very low data rate. Therefore, it can use a highly used channel. So, the user's qualified channels include all highly used, medium used, and almost free. If the user is using some live streaming applications or large file sharing with another user, then it needs an almost free channel. Therefore, the user's qualified channels include only almost free channels.

We assume that the CR BS can operate on multiple channels simultaneously. When a user moves to another location it can communicate with the CR BS on any permitted channel. While the user moves to another block and the channel it is using is no longer in its qualified channel list then it needs to switch to another channel. When a pair of users communicate directly and move together, then the channel they are operating on may not be available at other locations. In this situation, both users need to switch to another channel. Channel switching is a costly mechanism because the user needs to re-establish connection with the other end which takes some time and interrupts the normal data flow. The new channel is selected from the user's qualified channels in the next blocks. The user selects the channels in such a way that over the next few moves the expected total number of channel switches is minimum.

Let there be $N$ ($CH_0, CH_1, \ldots, CH_{N-1}$) unlicensed channels and $B$ ($0, 1, \ldots, B-1$) blocks. At time $t_0$ the user is at block $b_i$. The mobility predictor predicts that at time $t_1$ the user moves to blocks $b_j$ with a probability of $p_{ij}$. Let the expected number of channel switches at $b_j$ be $C(b_j, CH_n)$ if the user uses channel $CH_n$.

Then, the expected number of channel switches in block $b_i$ is expressed as the following:

$$C(b_i, CH_n) = \sum_{j=0}^{B-1} \left\{ p_{ij} \min_{\forall 0 \leq n' < N} \left( C(b_j, CH'_n) + X(n, n') \right) \right\} \tag{1}$$

Here, $X(n, n')$ is 1 (or 0) if $n$ is not equal to $n'$ ( or $n$ is equal to $n'$). $\left( C(b_j, CH'_n) + X(n, n') \right)$ this part of the Equation (1) indicates the number of channel switches from block $i$ to $j$. If the channel in $j$ is the same as the channel in the current location then the user does not need to switch the channel ($X(n, n') = 0$). In this case, the cost at block $i$ is the same as block $j$. We consider movement to all possible values of $j$. Therefore, the expected cost is a weighted sum of the number of channel switches of all possible next locations.

### 4.1. Problem Formulation

Let $CH_n$ be the channel selected by the user when its current location is block $b_0$. Therefore, the problem is to find a $CH_n$ that minimizes the expected channel switching cost. The problem can be expressed as the following:

$$\begin{aligned} \text{minimize} \quad & C(b_0, CH_n) \\ \text{subject to} \quad & \forall_{c \in UC(b_i)} Q(c) > R \end{aligned} \tag{2}$$

Here, $UC(b_i)$ is the usable channels in block $b_i$, $Q(c)$ is the quality of the channel, and $R$ is the quality required by the users. The constraint indicates that the quality of the selected channel cannot be lower than the required quality.

### 4.2. A Graph-Based Solution

To find the best channel, we first formulate a directed graph considering all possible transitions of a few next moves. From the directed graph we extract the transition trees. The number of transition

trees is equal to the number of qualified channels at the current location of the user. The qualified channels in current positions are considered as roots of the trees. The trees are identical except at the root level. Therefore, we need to calculate the cost once for every subtree at the root. Then, we pick the channel that produces the minimum cost. The complete algorithm is shown in Algorithm 1.

---

**Algorithm 1** Find the best channel

---

**Input:** The transition graph $G$, spectrum database $SD$, required channel quality.
**Output:** The best channel $ch$.
 1: **Procedure:** FIND-BEST-CHANNEL($G, SD, R$)
 2:    $T \leftarrow$ transition tree from $G$
 3:    Initialize $C[b_i, CH_N]$ for all leaf nodes.
 4:    **for** node $n \in T$ bottom-up order **do**
 5:       Calculate $C(b_i, CH_N)$ according to Equation (1).
 6:    **end for**
 7:    $ch \leftarrow$ ARGMIN($C(b_0, CH_N)$).
 8:    **return** $ch$.
 9: **end Procedure:**

---

Let us consider the example in Figure 3a. There are two channels: $CH_1$ and $CH_2$. User qualified channels in blocks 0, 1, 2 ,3 ,4, and 5 are $\{CH_1, CH_2\}$, $\{CH_2\}$, $\{CH_2\}$, $\{CH_2\}$, $\{CH_1\}$, and $\{CH_1, CH_2\}$, respectively. It is predicted that the user from block 0 moves to block 1 and block 4 with a probability of 0.7 and 0.3, respectively. From block 1, the user moves to block 2 and block 0 with a probability of 0.5 and 0.5, respectively. From block 4, the user moves to block 5 and block 3 with an equal probability. We extract the transition tree (shown in Figure 3b) for $CH_1$ from the graph. The cost of transition is 0 when there are no channel switches, otherwise it is 1.

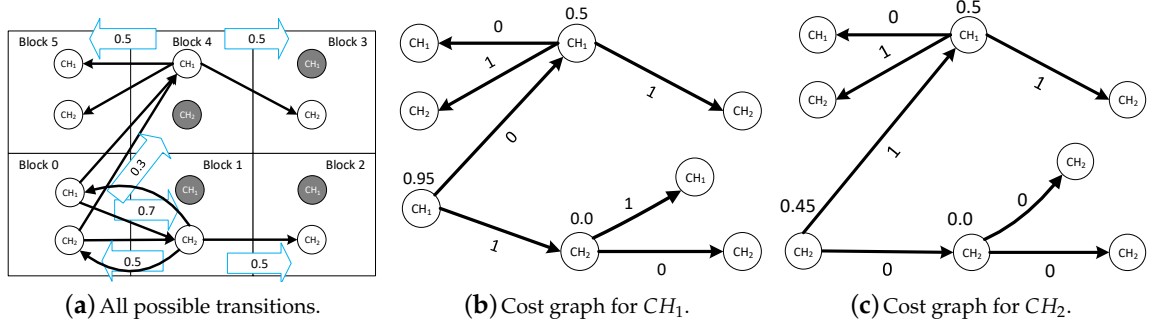

(**a**) All possible transitions.    (**b**) Cost graph for $CH_1$.    (**c**) Cost graph for $CH_2$.

**Figure 3.** Solution steps with an example.

Next, we calculate the minimum expected cost at each node in the transition tree. From $CH_1$ at block 4, the user can move to block 0 on $CH_1$ or $CH_2$. The minimum switching cost is 0. From $CH_1$ at block 4, the user can move to block 3 on $CH_2$ and the cost is 1. The minimum cost at block 4 for moving to block 3 is 1. Therefore, the minimum expected cost at block 4 is $0 \times 0.5 + 1 \times 0.5 = 0.5$. Similarly, we calculate the minimum expected cost at block 1 which is 0. Finally, we calculate the expected minimum cost of $CH_1$ at block 0 which is $(0.5 + 0) \times 0.3 + (0 + 1) \times 0.7 = 0.85$. Similarly, we calculate the expected minimum cost for $CH_2$ at block 0 which is $(0.5 + 1) \times 0.3 + (0 + 0) \times 0.7 = 0.45$. Therefore, the user selects $CH_2$.

## 5. Predicting Future Locations of Users

According to the system model, a user stores the block locations and time in their local storage. Each user predicts their next locations based on history. To predict the next location of a user, we first construct a mobility pattern database. A mobility pattern is a series of block location with time. We cut the trajectory of a user into parts and the parts are used as mobility patterns. Whenever a user remains at the same block for a certain period of time, we place a cut in that location in the trajectory data.

We denote the time as the waiting time. We also place a cut when the time difference between two consecutive blocks is more than the waiting time.

After getting the mobility patterns, we match the recent trajectory of a certain length of a user with the patterns. Then, the next block is predicted from the matched mobility pattern. To compare user trajectory with mobility pattern we calculate the distance between them. The distance between two trajectories is given by the following:

$$d(T, T') = \min_{\delta=0}^{L'-L} \left\{ \sum_{i=0}^{L} dist(T[i] - T'[i + \delta]) \right\} \tag{3}$$

Here, $L$ and $L'$ are the length of trajectory $T$ and $T'$, respectively. $dist(b_0, b_1)$ is the distance between two blocks. If $d(T, T')$ is smaller than a threshold $Th$, then it is considered as a match. It is possible that the user's recent trajectory data matches with multiple mobility patterns. In that case, multiple future locations are predicted and their probability is calculated from the distances of the patterns. If the predicted next locations are $b_1', b_2', \ldots, b_N'$ with a distance of $d_1, d_2, \ldots, d_N$, then the probability of selecting the next location is given by the following equation:

$$p(b_0, b_i') = \frac{1}{N} \frac{d_i'}{\sum d_i'} \tag{4}$$

There are some situations when the current location of the user is at the end of a mobility pattern. In this situation, it is hard to find which mobility pattern it is going to follow next. To overcome this problem, we calculate the probability of transition from one mobility pattern to another based on history. Then, we predict the next locations from those mobility patterns.

*Experimental Results*

We use the Microsoft GeoLife Trajectory [1] dataset for testing our mobility prediction model. The trajectory data is collected in Microsoft research area in a period of three years. The data contains GPS locations (latitude, longitude, and altitude) and timestamp of 182 users. This dataset contains 17,621 locations of a total distance of about 1.2 million km and a total duration of 48,000 h. These locations are recorded by different GPS loggers with different sampling rates. This dataset contains the outdoor movements of a broad range of users. It includes life routines (go home or go to work) and some entertainment and sports activities (shopping, sightseeing, dining, hiking, and cycling).

We first divide the trajectory of each user into train and test trajectories (60/40 ratio). Then we extract the mobility patterns from each user's train trajectories in the dataset. We keep the waiting time to be 25 min. Among the 182 users, we select the most 100 users by the amount of data. Then for each user, we create a separate mobility pattern database. Figure 4 shows some of the patterns of user 0. We can observe that some of the patterns are very random. Therefore, matching these patterns might not produce a good next location prediction. Next, we divide the test trajectories into some parts of different lengths. Then each part is matched with the patterns in the database and we select the patterns that are closest to the test trajectory part. We find the closest point on each mobility pattern from the last location of the part. Then, the next locations are selected from the closest point on the pattern. The expected time and probability are also calculated from the next locations and time. Figure 5 shows the accuracy of the next location prediction of each user. We can observe that some users show very good accuracy (greater than 80) and some of them are very low (below 10%). This is because some of the users may follow regular schedules and path while some of them are mostly random. Missing data of the users for some certain period of time is another reason for low accuracy. The average accuracy of all users is 33%. The median and 75 percentile accuracy is 32% and 50%, respectively.

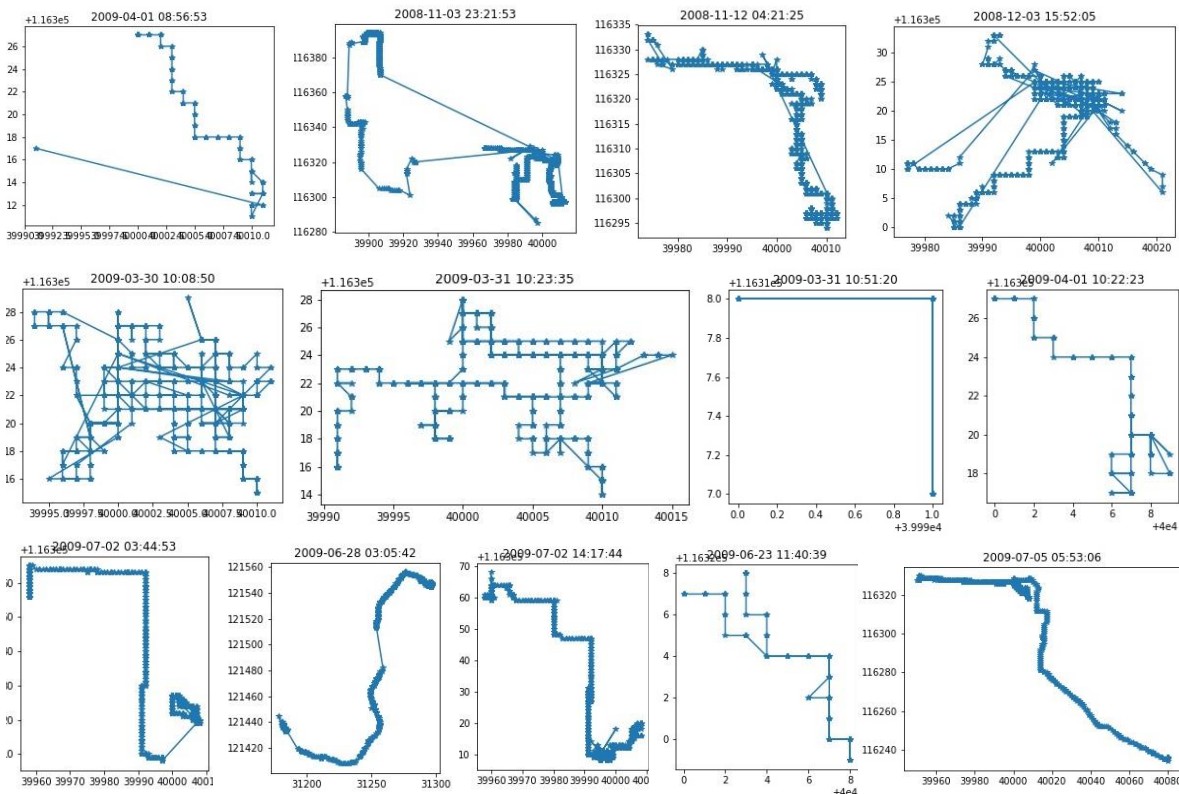

**Figure 4.** Mobility patterns of user 0.

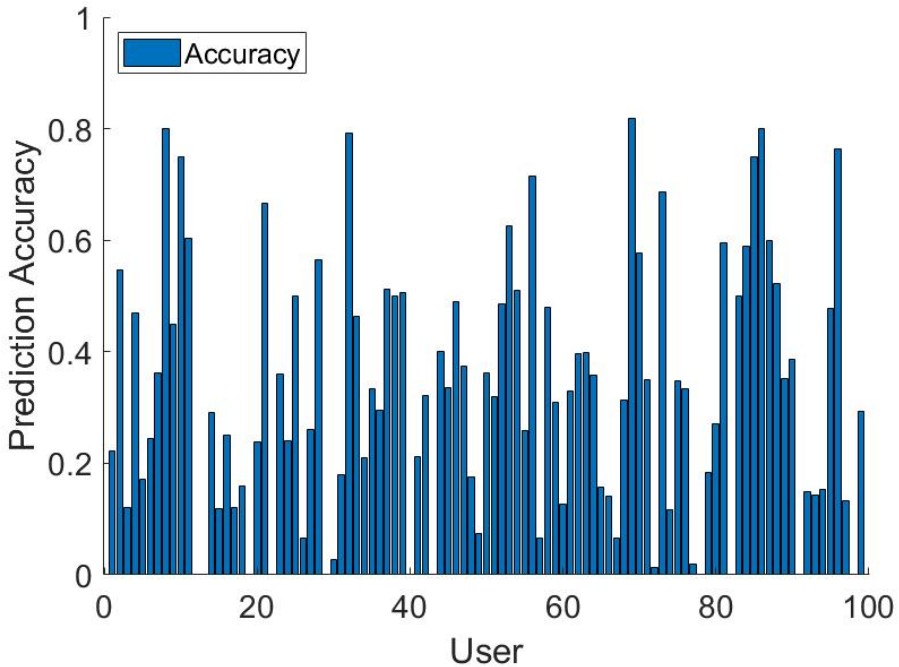

**Figure 5.** Accuracy of mobility prediction.

## 6. Obtaining Spectrum Database

At first, we divide the dataset area into 50 × 50 meter blocks. We assume that in each block the channel state will remain the same. Normally, the Wi-Fi transmission range is about 100 m in an outdoor environment. We first select the APs which are within 150 m from the center of a block.

Then we calculate the signal strength at the center for each channel. We keep the maximum signal strength on each channel and count the number of APs on that channel. To obtain signal strength, we use the free space path loss formula. The free space path loss formula is the following:

$$\frac{P_r}{P_t} = D_t D_r \left(\frac{c}{4\pi d f}\right)^2 \tag{5}$$

Here, $P_t$ and $P_r$ are the transmission power and received power in watt. $D_t$ and $D_r$ are the directivities of the transmitting and receiving antenna. We assume the antennas are isotropic and have no directivity ($D_t = D_r = 1$). $d$, $f$, and $c$ are the distance (m) between transmitter and receiver, frequency (Hz) of the transmitter, and speed of light ($3 \times 10^8$ m/s).

We also assume that all the AP are transmitting with the highest allowed power. According to FCC, the highest transmission power allowed in 2.4 GHz and some of the 5 GHz bands is 30 dBm which is equivalent to 1 watt. Therefore, we set $P_t = 1$ and calculate the $P_r$ from Equation (5). For example, let an AP operating in $CH_1$ be 20 m away from the center of the block. The center frequency of $CH_1$ is 2,412,000,000 Hz. Therefore, the received signal power will be $\frac{1\times4\pi20\times2.412\times10^9}{3\times10^8} = 2.44 \times 10^{-7}$ which is equal to $-66.10$ dBm. This way we calculate the signal power in from each AP within 150 m and store the highest value in our spectrum database. Therefore, our spectrum database is a four-tuple of location, signal power, and number of APs ($< latitude, longitude, signal\,power, number\,of\,APs >$).

### 6.1. Predicting Channel Status

We assume that collecting channel status information from each block might not be possible. All blocks might not be accessible to the public. It is also hard to collect channel status in each block. Therefore, we need to predict the channel status of some blocks from channel status information of the nearby blocks. There is a strong relationship between two adjacent blocks. Firstly, the signal from a block propagates to adjacent blocks. Secondly, the numbers of APs in a region are similar. For example, in a metropolitan area, most of the blocks will be highly dense with APs and most of the channels are congested. Secondly, an AP operates on the channel which is less congested. This is a kind of opposite relation with the nearby APs. Therefore, the relation between adjacent blocks is a mix of similarity and opposite. We use decision tree (DT), random forest (RF), artificial neural network (ANN), and support vector machine (SVM) approaches to predict channel status.

### 6.2. Features Used for Prediction

We use two properties from each channel: the number of APs and signal power at the center of a block. There are 11 ($CH_1 - CH_{11}$), 8 ($CH_{131} - CH_{138}$), and 30 unlicensed channels in 2.4 GHz, 3.65 GHz, and 5 GHz bands. Therefore, there are $49 \times 2$ features of a block. There can be four types of predictors based on the number of observed blocks around the block to be predicted. Therefore, the features of each channel are:

- **Number of APs:** The total number of reachable APs in a channel at the center of a block.
- **Signal power:** The maximum signal power in dBm in a channel at the center of a block.
- **Time:** The time in a coarse granularity (early morning (12:00 a.m.–6.00 a.m.), morning (6.00 a.m.–12.00 p.m.), afternoon (12.00 p.m.–6 p.m.), late afternoon (6.00 p.m.–12.00 a.m.)).

By using these features, we predict the number of APs and signal power in different methods.

### 6.3. Artificial Neural Network

The details of the ANN structure are shown in Figure 6. We have $49 \times 2 \times i = 98i$ inputs in the input layer, where $i$ ($1 \leq i \leq 4$) is the number of known blocks. We have two hidden layers with 49 and 98 neurons, respectively. We use the most commonly used activation function ReLU as the activation function in our ANN. This structure works well to find the number of APs and signal power. This two hidden layered ANN works much better than a one-hidden layered ANN.

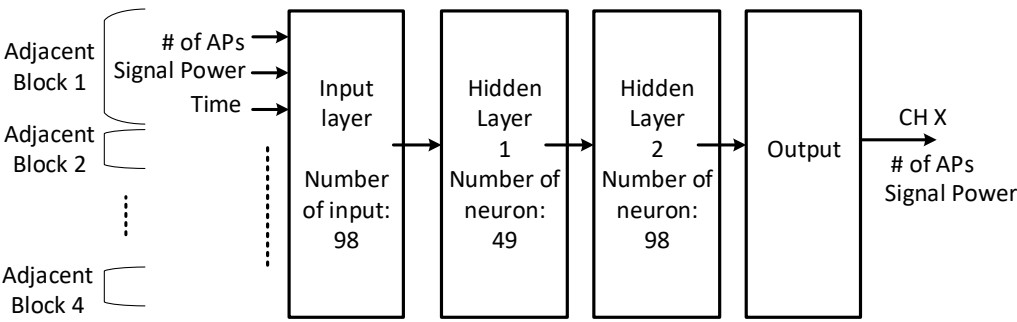

**Figure 6.** Structure of ANN.

### 6.4. Other Machine Learning Approaches

We also use Support Vector Machine, Decision Tree, K-nearest Neighbor, and Random Forest classifier to predict the channel status. We train the predictor for each channel using $49 \times 2 \times i = 98i$ input features. Therefore, for each channel, we have a separate classifier. The total number of classifiers in a classifier group is 49. We use decision tree, k-nearest neighbor, random forest, and support vector machine (linear and non-linear).

### 6.5. Experimental Result

We conduct an experiment to test the performances of the channel status predictors. We first collect the AP locations from WiGLE [17] dataset of 2000 m × 1000 m are at Center City, Philadelphia, PA. Figure 7 plots a part of the AP locations dataset. The dataset contains the locations of APs and their used channels. The dataset does not contain any channel usage information of the APs in different time. Therefore, we needed to change the channel of AP randomly with a probability of 0.01. At this point, we have a spectrum dataset of three dimensions: location, channel, and time. Figure 8 shows the signal powers and the number of APs in the morning for channel 1 (2.4 GHz). Figure 9 shows the signal powers and number of APs in the evening for channel 153 (5 GHz).

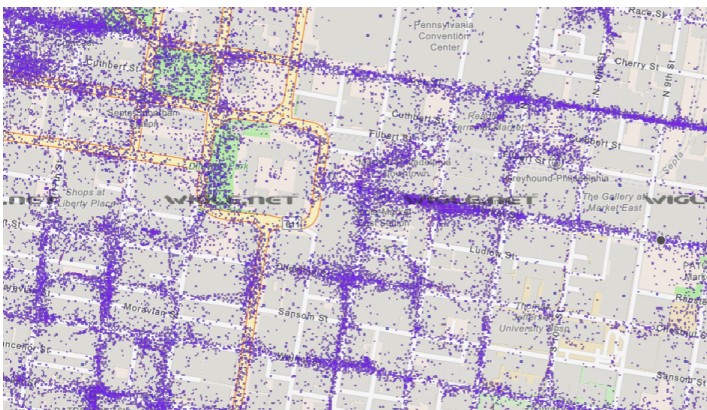

**Figure 7.** Wi-Fi dataset at Center City, Philadelphia, PA, USA. [17].

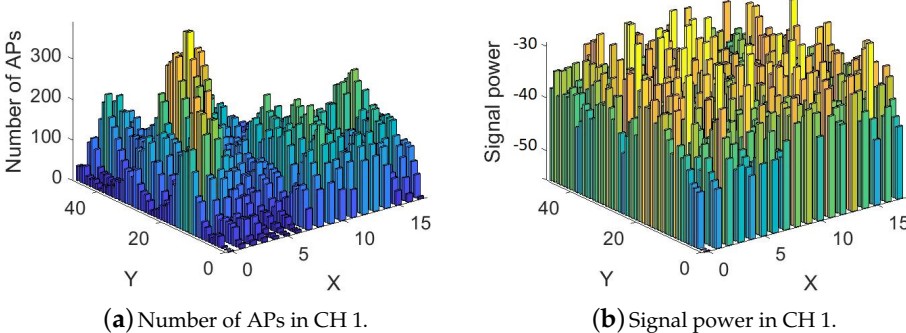

(**a**) Number of APs in CH 1.

(**b**) Signal power in CH 1.

**Figure 8.** Signal powers and APs in the morning.

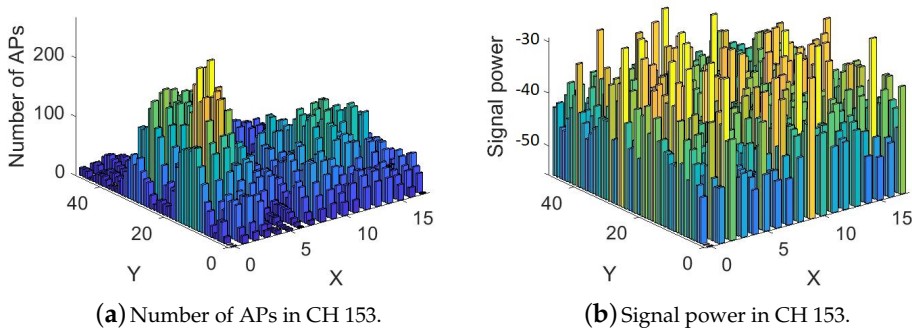

(**a**) Number of APs in CH 153.

(**b**) Signal power in CH 153.

**Figure 9.** Signal powers and APs in the afternoon.

Next, we split the dataset into train and test with a 60/40 ratio. We train the models mentioned earlier and test on the test dataset. There are some classifiers that cannot work on real numbers (DR, KNN, FR, SVM). Therefore, we categorize the signal strength. The step size for signal power and the number of APs are kept 2 and 10, respectively. For example, the blocks with the number of APs between 0 and 9 in a particular channel goes to class 1 and the number of APs between 10–19 goes to class 2. The error (average deviation from the actual signal strength in dBm) of predicting signal strength is shown in Figure 10a. We can observe that the random forest classifier can produce the lowest error among other classifiers. The error (average deviation from the actual number of APs) of predicting the number of APs is shown in Figure 10b. We can observe that the support vector machine (gamma) classifier can produce the lowest error among other classifiers.

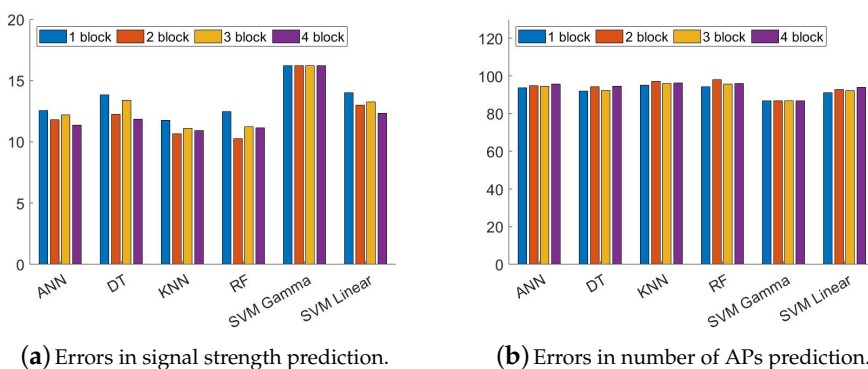

(**a**) Errors in signal strength prediction.

(**b**) Errors in number of APs prediction.

**Figure 10.** Errors in number of APs prediction.

## 7. Simulation

In this section, we describe our simulation settings and simulation results.

### 7.1. Simulation Settings

We conduct the simulation in a custom build java simulator. We implement the mobility prediction scheme and the spectrum database formulation and calculate the cost of channel switches. We compare the expected cost of the proposed channel selection mechanism with the actual switching cost if the channel is selected. It is hard to compare our approach with existing approaches because of different settings. Most of them do not consider future locations of the users. However, we compare with an existing approach [15] that selects the best channel. Though the existing approach considers multiple parameters, in our setting we can consider only the signal power. Therefore, we consider the channel with the lowest signal power as the best channel. We compare our approach with the minimum possible switching cost and cost for selecting the best channel.

We conduct simulation with two randomly generated AP datasets with different channel settings. Dataset 1 is sparse and contains 10,000 APs. Dataset 2 is highly dense with 40,000 APs. The APs are placed in the considered area uniform randomly. We first consider a rectangular region that covers all the GPS locations in the GeoLife trajectory dataset. Then the area is divided into multiple 50 m × 50 m blocks. A specific number of APs are placed at random locations in each block. Then the channel is assigned randomly. After channel assignment the signal strength and number of APs in each block are calculated. This spectrum dataset is considered as an early morning spectrum database. We change the channel randomly with a probability of 0.1 and store it as a morning spectrum database. Similarly, we obtain our afternoon, and late afternoon database. After that, we run the mobility prediction on the test dataset and calculate the transition graph and find the expected cost. We compare four measurements: expected cost, actual cost, best channel cost, and minimum cost. The expected cost is the expected number of channel switching according to Equation (1) when user follows the predicted trajectory. The actual cost is the number of channel switches if the channel is selected by our proposed approach (user follows the predicted trajectory) and the user follows the ground truth trajectory. The best channel cost is the is the number of channel switches if the channel with lowest signal power is selected. If there are multiple channels with the lowest signal power, then one of them is randomly selected. The minimum cost is the minimum number of channel switches if the user follows the ground truth trajectory.

### 7.2. Simulation Result

Figure 11a shows the expected cost, the actual cost, the best channel cost, and the lowest cost in dataset 1 for different number of channels available for usage. We vary the number of channels from 3 to 20 and keep the future location prediction up to three blocks. We can observe that the costs are reduced by the number of channels. The actual cost is always higher than the expected cost. This is because our mobility predictor is not perfect. The locations it predicts and the locations the user actually follow are different as well as the channel status. Therefore, the selected channel for predicted locations might cause extra channel switch. The cost for selecting the best channel remains almost constants with the number of channels. The actual cost is always lower than the best channel cost. We observe that the average number of channel switches in our proposed approach is 0.22 higher than that in the actual scenario which means in a movement of $3 \times \frac{1}{0.22} \approx 13$ blocks, a user might switch channels one more time than the minimum. The average number of channel switches for selecting the best channel is 0.32 higher than that in the actual scenario which means in a movement of $3 \times \frac{1}{0.32} \approx 9$ blocks, a user might switch channels one more time than the minimum. The minimum number of channel switches is similar to the expected number of channel switches.

Figure 11b shows the expected cost, actual cost, the best channel cost and the lowest cost in dataset 1 for different prediction levels. We vary the number of future prediction blocks from 2 to 6

and keep the number of channels as 5. We can observe that the costs increase with the prediction level. The actual cost is always higher than the expected cost as before. We observe that the average number of channel switches in our approach is 0.21 higher in the actual scenario. The average number of channel switches when the best channel is selected is 0.31 higher in the actual scenario. The minimum number of channel switches is a little higher than the expected number of channel switches.

Figure 11c shows the expected cost, actual cost, the best channel cost, and lowest cost in dataset 2 for different number of channels. We vary the number of channels from 3 to 20 and keep the number of future blocks as 3. We can observe that the costs decrease with the number of channels. The actual cost is about 0.28 higher than the expected cost as before. Therefore, in a movement of $3 \times \frac{1}{0.28} \approx 11$ blocks, a user might switch channels one more time than the minimum. The average number of channel switches for selecting the best channel is 0.41 higher than that in the actual scenario which means in a movement of $3 \times \frac{1}{0.41} \approx 7$ blocks, a user might switch channels one more time than the minimum. The overall cost is higher than that of dataset 1. This is because dataset 2 is denser than dataset 1. Therefore, the available user qualified channels are limited. The minimum number of channel switches is slightly higher than the expected number of the channel switches.

Figure 11d shows the expected cost, the actual cost, the best channel cost, and the lowest cost in dataset 2 for different prediction levels. We vary the number of future prediction blocks from 2 to 6 and keep the number of channels as 5. We observe a similar behavior to dataset 1. The average number of channel switches is 0.22 higher than that of the actual scenario. The minimum number of channel switches is a little higher than the expected number of channel switches. The average number of channel switches when the best channel is selected is 0.30 higher than that of the actual scenario. The minimum number of channel switches is also higher than the expected number of channel switches.

Therefore, we can conclude that the proposed channel selection approach works well enough even with a lower accuracy of the mobility prediction system. It is always better to select the channel with the lowest expected cost than the best channel.

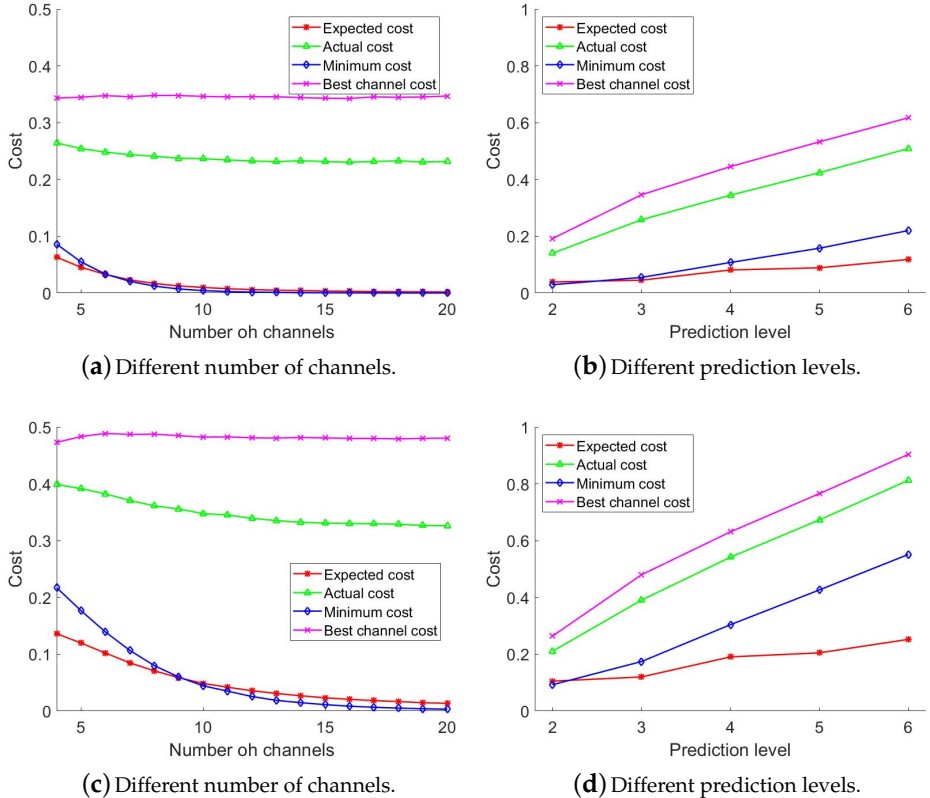

(**a**) Different number of channels.

(**b**) Different prediction levels.

(**c**) Different number of channels.

(**d**) Different prediction levels.

**Figure 11.** Simulation result.

## 8. Conclusions

The channel selection in cognitive radio plays an important role especially when the users are mobile, and the channel status is different within a short distance. The proper channel selection can reduce the number of channel switches while the user moves to another place. In this work, we propose a mechanism to select channels from the available channels that meet user's requirement. We calculate the expected channel switching costs over the channels that meet users' requirements. The expected channel switching cost is based on the predicted locations, probability of move, and the channel status of the location at the arrival time. We construct a spectrum database containing the channel quality of location and time in a coarse granularity. We also apply a mobility prediction scheme based on mobility pattern matching. Our simulation results show that the actual channel switching cost is always lower than the best channel selection cost and slightly higher than the minimum cost with a low accuracy mobility predictor. In a trip of half kilometers, (11–13 blocks) a user may switch channels one more time than the minimum.

**Author Contributions:** Conceptualization, R.B. and J.W.; methodology, R.B.; software, R.B.; validation, R.B.; formal analysis, R.B.; investigation, R.B.; resources, R.B.; data curation, R.B.; writing–original draft preparation, R.B.; writing–review and editing, J.W.; visualization, R.B.; supervision, J.W.; project administration, J.W.; funding acquisition, J.W. All authors have read and agreed to the published version of the manuscript.

**Funding:** This research was supported in part by NSF grants CNS 1824440, CNS 1828363, CNS 1757533, CNS 1618398, CNS 1651947, and CNS 1564128.

**Conflicts of Interest:** We declare that there is no conflict of interest.

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
