# Peer review of "Minimizing The Number of Channel Switches of Mobile Users in Cognitive Radio Ad-Hoc Networks"

_jsan, doi:10.3390/jsan9020023_

Round 1

Reviewer 1 Report

Dear authors, the paper you proposed is really interesting in its goals and well describes the problems you can have when using CRAHN in ISM bands. The materials and methods are also adequately described. Unfortunately, the whole experimental part is hard to follow, it has several problems and does not provide good results in terms of obtained performance. 

Firstly, in the introduction (line 33), the expression "less than -30 dBm is considered as the best signal" is not correct. High powers are higher than -30 dBm, i.e. -20, -10 and so on. Therefore, "less than" can be used if dealing with the absolute value (20 is actually less than 30) but not if considering the full number. Furthermore, measurement units must be placed after a blank space, according to SI rules. The values -30, -50, -70, -90 are used as an example and it must be specified because there are technologies that are able to demodulate also -100 dBm signals (just think to DSS modulations).

Figure 3 is placed very far from the point it is cited in the text. Please consider replacing. 

Figure 4 is never cited in the text and it must be commented.

The accuracy obtained at line 306 in mobility prediction is quite low. Is the prediction method inaccurate or the user's randomness that cannot be reduced?

Figure 9 cannot be understood because of the Z axis whose values are unclear (all 0s in two cases, and .2 .4 in the other two cases).

Figure 10 reports the errors but they are absolute values, they cannot be really evaluated if not expressed in relative mode.

Figure 11 and the definition of costs must be reworked.

Conclusion do not fully support the results, because these last ones are not fully clear.

I really think that this paper deserves to be published but revisions should be accomplished otherwise it remains a very good idea, very well designed but poorly actuated in terms of results.

Reviewer 2 Report

The grammar and format must be improved.  Some cases are page 4, line 156, formatting on page 5, lines 157 to 160,

line 157 situation needs to be changed to situations, etc.  There are many instances in the paper where the work switch is used, however switches should have been used because the plural case was intended.

One page 5, the statement is not correct that TV transmission ranges from 100 miles to 200 miles.  In fact, a typical

television transmission distance limitation is 60 to 80 miles, depending on the height of the transmit antenna

and the transmit power. 

On page 6, equation (1) needs additional explanation and some correction.  There should be brackets to include

the X(n,n') term in the minimization.  In line 246, pic should be changed to pick.

Figure 4 is much too small to be readable.  This needs to be improved.

On page 9, equation 5 is incorrect.  Equation 5 implies that the received power increases with the distanced squared.

This is incorrect.  The left hand side of the equation should be Pt/Pr instead of Pr/Pt. 

I would argue that this is the wrong signal propagation model entirely.

A two-ray propagation model should have been used such that the Pr decreases with distance

to the forth power instead of distance squared. All of the numerical simulation in the paper should be redone with the two-ray propagation  model instead of the free-space propagation model.

Round 2

Reviewer 1 Report

The authors have addressed most of my comments. There is still a crucial issue: their performance level is not satisfactory. In this state of things, there is a comparison between their proposal and the minimum required cost and their approach is always worse. In my opinion, they should compare their approach with an existing one and prove that they have improved somehow those performance. Otherwise, although they propose an integrated system, there are more reason to avoid publishing than allowing it.

Round 3

Reviewer 1 Report

The authors have correctly addressed all my comments.